# The association between bullying and toothache in Brazilian students: An analysis of the Brazilian National Student Health Survey

Caroline Segatto Girardon[1], Maria Laura Braccini Fagundes[1], Fernando Neves Hugo[2]*,
Jessye Melgarejo do Amaral Giordani[1], Luana Severo Alves[1], Orlando Luiz do Amaral Júnior[1]

**1** Postgraduate Program in Dental Sciences, Federal University of Santa Maria, Santa Maria, Brazil,
**2** College of Dentistry, Department of Epidemiology and Health Promotion, New York University, New York, New York, United States of America

* fnh9064@nyu.edu

## Abstract

This study analyzed the association between self-perceived bullying and self-reported toothache among Brazilian students and evaluated the moderating role of school-based health actions, including participation in the School Health Program, oral health promotion, and bullying prevention. A cross-sectional study was conducted using data from the 2019 National School-based Health Survey, including 53,711 students aged 13–17 years. The outcome was self-reported toothache and the main exposure was self-perceived bullying. Moderating variables included school participation in the School Health Program, oral health promotion actions, and bullying prevention actions. Poisson regression models with robust variance were fitted, with standard errors adjusted for clustering by school. Overall, 23.6% of students reported toothache and 13.7% reported bullying. Moderation analyses showed no evidence that school health actions influenced the association between bullying and toothache. For bullying once or ≥2 times, prevalence ratios were: School Health Program participation (PR_once = 0.92, 95%CI 0.77–1.09; PR_≥2 = 1.07, 95%CI 0.81–1.41), bullying prevention (PR_once = 0.98, 95%CI 0.76–1.25; PR_≥2 = 0.82, 95%CI 0.72–1.09), and oral health promotion (PR_once = 1.13, 95%CI 0.96–1.33; PR_≥2 = 1.19, 95%CI 0.94–1.52). These findings indicate that school-based health actions alone may be insufficient to mitigate the impact of bullying on adolescents' oral health.

### Introduction

The health of children and adolescents is influenced by social determinants, which encompasses living conditions, access to resources, and daily experiences within institutional settings, such as the school environment [1]. Among these determinants, school bullying, defined as a form of interpersonal violence characterized by repeated aggression and power imbalance between victims and aggressors, has been

**Data availability statement:** This study exclusively utilized secondary, anonymized data from the 2019 edition of the National School Health Survey (Pesquisa Nacional de Saúde do Escolar – PeNSE), a publicly available dataset provided by the Brazilian Institute of Geography and Statistics (Instituto Brasileiro de Geografia e Estatística – IBGE) in partnership with the Ministry of Health. All data are aggregated and anonymized, ensuring the protection of individual privacy in accordance with national ethical standards. As publicly accessible data collected through a governmental initiative, the use of this dataset does not require additional ethical approval. For more information on data access and ethical considerations, please visit the IBGE website: https://www.ibge.gov.br/estatisticas/sociais/saude/9134-pesquisa-nacional-de-saude-do-escolar.html.

**Funding:** The authors received no specific funding for this work. The Article Processing Charge (APC) was covered through an institutional agreement between New York University and PLOS Global Public Health. The funder had no role in study design, data collection and analysis, decision to publish, or preparation of the manuscript.

**Competing interests:** The authors have declared that no competing interests exist.

extensively studied due to its negative effects on the physical and mental health of students [2]. Its recognition as a social determinant of health demonstrates the need for approaches to address multiple forms of violence across school ecosystems [3].

Epidemiological data indicate that toothache affects a considerable portion of the population, especially schoolchildren, being frequently reported in this group [4]. Furthermore, studies have shown that Brazilian adolescents who experience bullying on a daily basis exhibit worse health-related indicators, which include not only toothache but also orofacial trauma, both of which negatively impact academic performance [5]. These findings suggest the existence of mechanisms linking these negative experiences to oral health outcomes. Among these mechanisms, chronic psychosocial stress stands out, as it may amplify pain perception and the impact of mental health on self-care [6], and social stigma, which affects access to services and participation in health promotion activities [7].

In Brazil, the School Health Program (Programa Saúde na Escola - PSE) is an intersectoral strategy, promoted by the Brazilian Ministries of Health and Education, which aims to integrate health and education policies and actions to promote the health and quality of life of students in the public education system. The PSE has shown potential to improve oral health [8] by integrating health education, disease prevention, and health promotion strategies. Rigorous and continuous evaluation of program implementation is essential for optimizing shared governance, informing the allocation of resources, and assessing its effectiveness on complex epidemiological phenomena, such as the association between bullying experiences and toothache.

In this context, the aims of this study were to analyze the association between self-perceived bullying and self-reported toothache among Brazilian students, and to evaluate the moderating role of three school-based health program actions: participation in the PSE, oral health promotion, and bullying prevention. Within this framework, school-based actions promoted by the PSE, including oral health initiatives and bullying prevention strategies, are expected to act as contextual protective factors. By emphasizing this relationship, the study contributes to the understanding of how psychosocial adversity influences oral health outcomes and to the critical evaluation of existing school health strategies.

## Methodology

### Design and scenario

This study used data from the National School-based Health Survey (PeNSE), conducted by the Brazilian Institute of Geography and Statistics (IBGE) in collaboration with the Ministry of Health. Before data collection, students were informed about the survey's objectives, the voluntary nature of their participation, and their right to withdraw at any time. Those who consented completed a structured, self-administered questionnaire on a smartphone, supervised by IBGE personnel. The questionnaire addressed various domains, including socioeconomic status, family environment, use of tobacco, alcohol, and other drugs, experiences of violence, accidents, perceptions of safety, and other aspects of adolescent life [9].

The sample comprised students aged 13–17 years enrolled in public elementary and high schools. The sampling design ensured representativeness at the national, regional, state, and municipal levels (including state capitals and the Federal District). A two-stage cluster sampling method was used, with schools as the primary sampling units and classrooms as the secondary units. All students in the selected classrooms were invited to participate. Data were collected from 4,242 schools and 6,612 classrooms, totaling 159,245 students. The overall non-response rate was 15.4% in 2019. Additional methodological details are available in technical reports published by IBGE [9].

## Ethical aspects

The PeNSE was approved by the National Research Ethics Commission of the Brazilian Ministry of Health (CONEP/MS), under Certificate of Ethical Approval (CAAE) No. 3.249.268, and complies with the guidelines and regulatory standards for research involving human subjects. The PeNSE microdata and respective documentation are publicly available from the Brazilian Institute of Geography and Statistics (IBGE) website (https://www.ibge.gov.br/estatisticas/sociais/educacao/9134-pesquisa-nacional-de-saude-do-escolar.html?=&t=microdados).

## Variables

**Outcome.** The outcome of this study was the experience of toothache in the past six months. This variable was assessed through the question: *"In the past six months, have you had a toothache that was not caused by the use of braces?"* with the response options "yes" and "no."

**Main predictor.** The main predictor was bullying experience, which was assessed by the following question: *"In the past 30 days, how many times have your schoolmates mocked, teased, bullied, intimidated, or ridiculed you to the point that you felt hurt, upset, annoyed, offended, or humiliated?"* Responses were categorized as none, once, or twice or more. [10].

**Moderating variables (contextual, school-level).** Three moderating variables were included in the analysis: whether the school participates in the PSE; whether the school carried out actions related to oral health promotion and assessment; and whether the school implemented bullying prevention actions. These variables were assessed through the following questions:

1) *"Does the school participate in the School Health Program (PSE)?"* (yes or no);

2) *"In the past 12 months, which of the following actions has the school undertaken? – Promotion and assessment of oral health?"* (yes or no);

3) *"In the past 12 months, which of the following actions has the school undertaken? – Prevention of bullying practices within the school premises?"* (yes or no).

**Covariates.** Covariates included for model adjustment comprised sociodemographic characteristics and health-related behaviors. The variables were area of residence (urban or rural); sex (male or female); self-reported skin color (white or brown/black); maternal education level (categorized as up to eight years of schooling or more than eight years). The selection of covariates for adjustment followed a causal framework based on directed acyclic graphs (DAGs) (Fig 1), in which variables representing oral hygiene practices, sugary beverage consumption, and mental health indicators were conceptualized as potential mediators on the pathway linking bullying to toothache. Therefore, these variables were not included in the main adjusted models to avoid overadjustment and preserve the estimation of the total effect of bullying on the outcome [11].

## Statistical analysis

Descriptive analyses were conducted using STATA 14.0 (Stata Corporation, College Station, TX, USA). Poisson regression models were estimated to investigate the association between self-perceived bullying and self-reported toothache.

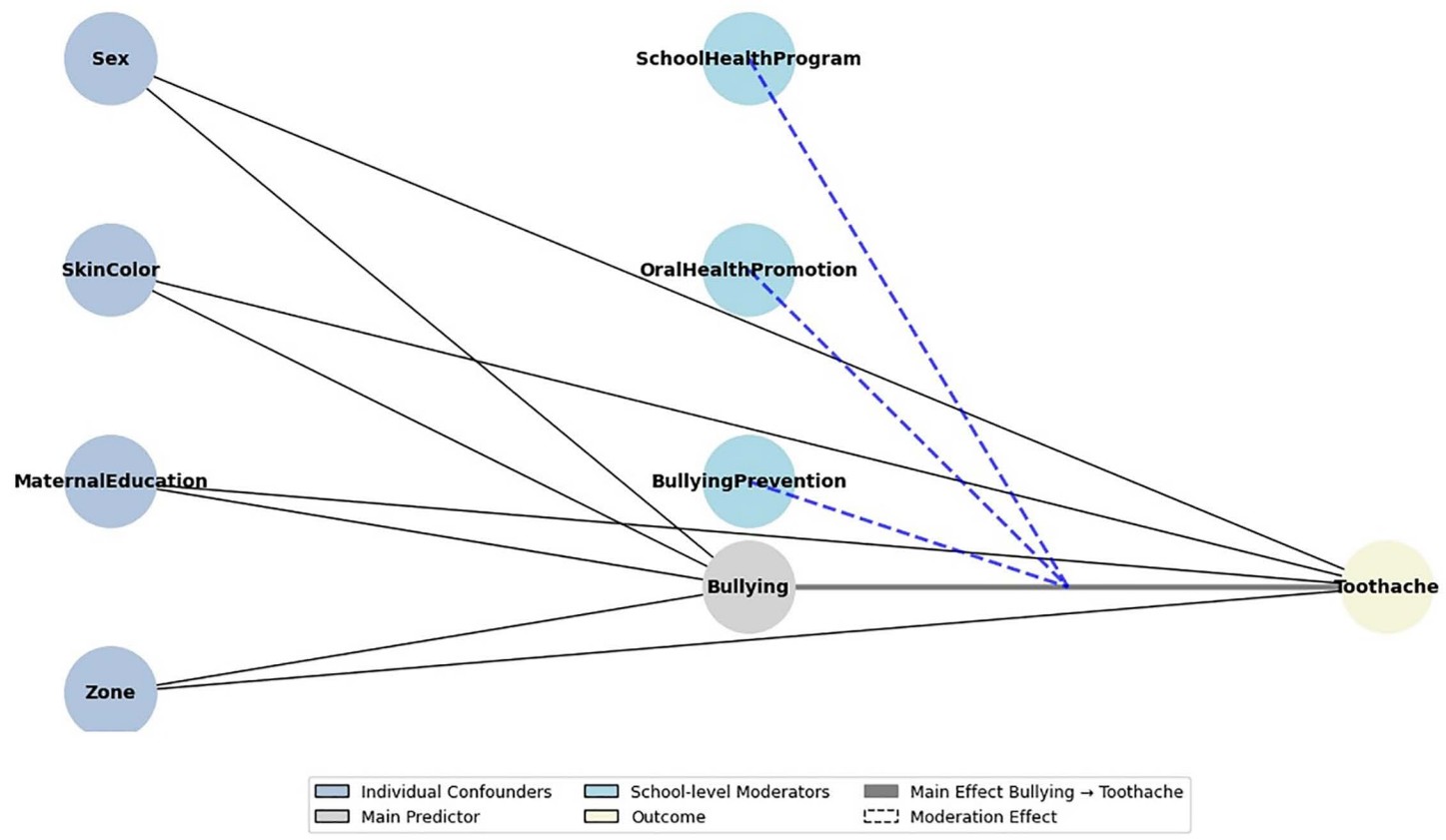

**Fig 1. Directed acyclic graphs (DAG) the depicting the association between bullying and toothache.** Solid arrow: main effect; dashed lines: moderation.

In addition, Poisson regression models with interaction terms were estimated to test the moderation hypothesis, aiming to understand the moderating effect variables on the association between perceived bullying and toothache.

Initially, data completeness was assessed. Detailed information on missing data patterns and comparisons between included and excluded participants is provided in the Appendix 1. A variable indicating complete information across all analytical variables was created (*completo2 = !missing(varlist)*), and analyses were restricted to these complete records to ensure data integrity and comparability across models, minimizing potential bias due to inconsistent missingness patterns and ensuring inclusion of the clustering variable (school) in all models. This step aimed to guarantee data integrity and internal consistency and was motivated by challenges faced when estimating multilevel Poisson models using the *mepoisson* command, which requires complete information at both individual and contextual levels. Initial attempts to fit multi-level outcome models were affected by convergence failures and numerical instability. In addition, clustering diagnostics indicated negligible between-school variability in the outcome, with an intraclass correlation coefficient (ICC) close to zero. Taken together, these findings indicated that multilevel outcome modeling would not provide additional inferential benefit. Therefore, population-averaged Poisson regression models with robust standard errors clustered at the school level were adopted. Sampling weights were applied to preserve the representativeness of estimates according to PeNSE's complex design.

Although this approach does not explicitly model between-school variance, it accounts for intraclass correlation by adjusting standard errors, thereby preserving the inferential validity of the estimated parameters. Importantly, even under

this alternative strategy, the selection of complete records was retained to ensure model comparability and to minimize bias arising from inconsistent missing data patterns [12].

The final analytical strategy involved testing interaction terms between individual-level variables (e.g., exposure to physical aggression) and each of the school-level moderating variables under investigation (PSE participation, bullying prevention, and oral health promotion actions). Models were estimated separately for each moderator, and in all cases, the remaining contextual variables were included as covariates to control for potential confounding effects. Three distinct models were tested, each including an interaction term between perceived bullying and one of the following: (1) participation in the PSE, (2) bullying prevention actions carried out by the school, and (3) oral health promotion actions carried out by the school. The inclusion of these interaction terms aimed to investigate potential moderating effects, defined as changes in the magnitude of the association between perceived bullying and toothache, depending on exposures at the school level. Main and interaction effects were expressed as prevalence ratios (PRs) with their respective 95% confidence intervals (CIs). A 5% significance level was adopted for all analyses. Although PeNSE adopts a complex sampling design, survey procedures using the svy prefix were applied only in the descriptive analyses to account for sampling weights, primary sampling units, and strata. For the regression analyses, Poisson models were estimated using sampling weights and robust standard errors clustered at the school level. Full survey-design–based estimation for the regression models was explored; however, due to technical constraints related to missing contextual identifiers and model stability in models including interaction terms, this approach was not adopted in the final analyses. Therefore, the regression estimates should be interpreted as population-averaged associations.

The DAG presented in Fig 1 was generated in Python using matplotlib and networkx. In addition, adjusted predicted prevalence of toothache by bullying frequency was graphically represented (Fig 2), and moderation of this association by school-level actions was also illustrated (Fig 3), including 95% CIs.

## Results

The sample consisted of 53,711 students. Table 1 presents the sample description, showing that 23.6% reported toothache in the past six months. Most participants lived in urban areas (91.5%), and the sex distribution was balanced, with 48.9% identifying as male. Regarding skin color, 64.6% self-identified as Black or Brown. Most mothers (60.9%) had more than eight years of formal education. With respect to bullying perception, 13.7% reported having experienced at least one episode in the past 30 days. Approximately half of the schools (50.3%) participated in the PSE, and 41.2% implemented anti-bullying actions. In addition, 45.9% of the schools implemented school-based activities focused on oral health promotion. Table 1 also presents the proportions of toothache occurrence in the past six months according to the independent variables. A higher prevalence of toothache was identified among female students (25.7%; 95%CI: 24.9–26.5) compared to males (19.7%; 95%CI: 18.9–20.5). Differences were also observed in relation to bullying: among those who reported being bullied once, the prevalence was 32.5% (95%CI: 30.1–35.1), and among those who reported two or more episodes, 28.6% (95%CI: 26.2–31.1), both higher than among those who did not report bullying episodes (21.5%; 95%CI: 20.9–22.1). These differences were observed based on the non-overlapping confidence intervals.

In Table 2, the prevalence ratios were close to the null for all comparisons: students in schools that did participate in the School Health Program had a similar prevalence of toothache compared to those in no participating schools (PR = 0.99; 95%CI: 0.91–1.08). No significant association was found for bullying prevention actions (PR = 0.97; 95%CI: 0.91–1.07). In contrast, oral health promotion activities showed a modest positive association with toothache reports (PR = 1.08; 95%CI: 1.01–1.16).

Table 3 presents the moderating effects of three school-level variables on the association between perceived bullying and toothache, assessed through interaction terms in Poisson regression models. Model 1 tested the interaction between bullying frequency and school participation in the School Health Program (PSE). The reference category comprised students who reported never being bullied and attended schools participating in the PSE. The confidence intervals for the

**Fig 2. Adjusted prevalence of toothache by bullying frequency.**

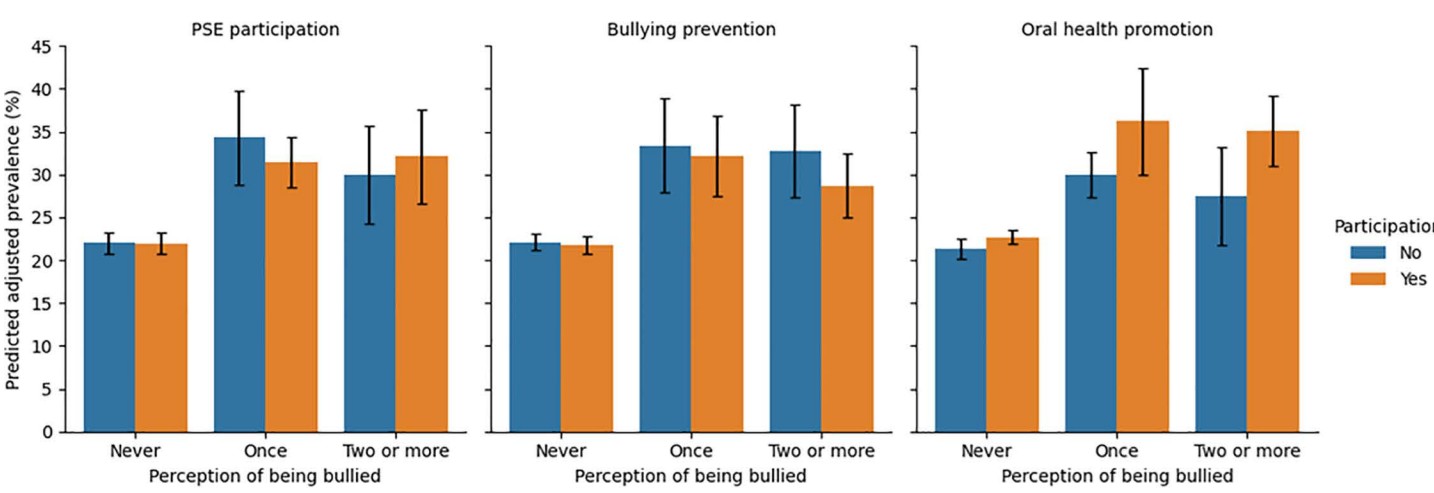

**Fig 3. Predicted adjusted prevalence (%) of perceived bullying on toothache moderated by school actions.**

**Table 1. Sample distribution and prevalence of toothache according to sociodemographic characteristics, bullying experience, and school-level variables (n = 53,711).**

| Variables | Weighted % | Prevalence of individuals with toothache by category (95% CI[a])* |
|---|---|---|
| **Sociodemographic** | | |
| *Zone* | | |
| Urban | 91.2 | 23.1 (22.4–23.7) |
| Rural | 08.7 | 19.7 (18.2–21.3) |
| *Sex* | | |
| Male | 48.9 | 19.7 (18.9–20.5) |
| Female | 51.0 | 25.7 (24.9–26.5) |
| *Skin color* | | |
| White | 34.9 | 22.6 (21.5–23.8) |
| Brown/Black | 65.0 | 22.8 (22.1–23.5) |
| *Maternal education level* | | |
| Up to 8 years of education | 42.6 | 24.0 (23.1–24.9) |
| More than 8 years of education | 57.3 | 22.4 (21.5–23.3) |
| *Frequency of bullying episodes in the last 30 days* | | |
| Never | 85.9 | 21.5 (20.9–22.1) |
| Once | 7.6 | 32.5 (30.1–35.1) |
| Twice or more | 6.3 | 28.6 (26.2–31.1) |
| *Schools that participate in the PSE* | | |
| Participate | 50.3 | 22.2 (21.5–23.0) |
| Don´t participate | 49.6 | 23.4 (22.5–24.3) |
| *Schools that carry out bullying prevention actions* | | |
| No | 59.0 | 22.7 (21.9–23.4) |
| Yes | 40.9 | 22.9 (22.0–23.8) |
| *Schools that carry out actions to promote oral health* | | |
| No | 53.9 | 21.8 (21.1–22.6) |
| Yes | 46.0 | 23.9 (23.0–24.8) |

[a] CI, confidence interval.

* Taking into account the sample weight.

interaction terms for bullying experienced once (PR = 0.92; 95% CI: 0.77 to 1.09) and twice or more (PR = 1.07; 95% CI: 0.81 to 1.41) included the null value, suggesting no evidence of moderation by PSE participation.

Model 2 assessed the potential moderating role of school bullying prevention actions. The reference group included students never bullied attending schools without bullying prevention activities. Interaction terms for bullying experienced once (PR = 0.98; 95% CI: 0.76 to 1.25) and twice or more (PR = 0.82; 95% CI: 0.72 to 1.09) yielded confidence intervals that indicate no significant interactons.

Model 3 examined the interaction between bullying and school-based oral health promotion actions. The reference group consisted of students never bullied attending schools that performed oral health promotion activities. The interaction terms for bullying once (PR = 1.13; 95% CI: 0.96 to 1.33) and twice or more (PR = 1.19; 95% CI: 0.94 to 1.52) also no significan interactions. In addition, Fig 2 shows the adjusted prevalence of toothache according to bullying frequency, with higher prevalence observed among students reporting bullying once or more. Fig 3 illustrates the moderating effects of school health actions on the association between bullying and toothache, indicating small or no meaningful modification.

**Table 2. Unadjusted and adjusted prevalence ratios (PR) for toothache according to individual and school-level characteristics. PeNSE, Brazil. (n = 53,711).**

| Variables | Toothache "Unadjusted analysis PR$^a$ (95% CI$^b$)*" | Toothache "Adjusted analysis PR$^a$ (95% CI$^b$)*" |
|---|---|---|
| **Sociodemographic** | | |
| *Zone* | | |
| Urban | 1 | 1 |
| Rural | 0.83 (0.75–0.91)** | 0.86 (0.70–0.99)** |
| *Sex* | | |
| Male | 1 | 1 |
| Female | 1.30 (1.23–1.38)** | 1.31 (1.24–1.38)** |
| *Skin color* | | |
| White | 1 | 1 |
| Brown/Black | 0.99 (0.93–1.05) | 0.99 (0.91–1.08) |
| *Maternal education level* | | |
| Up to 8 years of education | 1 | 1 |
| More than 8 years of education | 0.95 (0.89–1.01) | 0.95 (0.90–1.01) |
| *Frequency of bullying episodes in the last 30 days* | | |
| Never | 1 | 1 |
| Once | 1.50 (1.37–1.65)** | 1.49 (1.34–1.65)** |
| Twice or more | 1.36 (1.23–1.51)** | 1.40 (1.34–1.56)** |
| *Schools that participate in the PSE* | | |
| Don´t participate | 1 | 1 |
| Participate | 0.94 (0.89–1.00) | 0.99 (0.91–1.07) |
| *Schools that carry out bullying prevention actions* | | |
| No | 1 | 1 |
| Yes | 0.98 (0.93–1.05) | 0.97 (0.91–1.07) |
| *Schools that carry out actions to promote oral health* | | |
| Yes | 1 | 1 |
| No | 1.07 (1.01–1.13) | 1.08 (1.01–1.16)** |

$^a$ PR, prevalence ratio; $^b$CI, confidence interval. Adjusted for socioeconomic and demographic factors.

* Taking into account the sample weight.

** $p < 0.005$.

## Discussion

The association between bullying experience and dental pain reflects the complex interplay between psychosocial factors and oral health outcomes. In addressing the specific pattern noted in our results, where students who were bullied once reported a slightly higher prevalence of toothache than those bullied multiple times, we propose that this may be explained by the heightened psychological impact of an initial, salient episode or by the development of coping mechanisms among recurrent victims. Bullying, as a chronic social stressor, can contribute to the onset or exacerbation of oral diseases through biological and behavioral pathways, such as increased inflammation, neglect of oral hygiene, and heightened pain perception. Dental pain, often a proxy for untreated oral conditions, may therefore be influenced not only by biological factors but also by social determinants like bullying. Understanding this relationship, including its nuanced patterns, is crucial for developing comprehensive strategies that address both the psychosocial context and the physical health of adolescents.

**Table 3. Moderating effects of School Health Program (PSE) participation, bullying prevention, and oral health promotion actions on the association between perceived bullying and toothache among Brazilian students (n = 53,711).**

| Moderating effects | PR[a] (95% CI[b])* |
|---|---|
| **(Model 1) Perception of being bullied moderated by PSE participation** | |
| Never##participate in the PSE | 1 |
| Once##participate in the PSE | 0.92 (0.77–1.09) |
| Two or more times##participate in the PSE | 1.07 (0.81–1.41) |
| **(Model 2) Perception of being bullied moderated by bullying prevention actions at school** | |
| Never##perform actions | 1 |
| Once##perform actions | 0.98 (0.76–1.25) |
| Two or more times##perform actions | 0.82 (0.72–1.09) |
| **(Model 3) Perception of being bullying moderated by oral health promotion actions at school** | |
| Never##perform actions | 1 |
| Once##perform actions | 1.13 (0.96–1.33) |
| Two or more times##perform actions | 1.19 (0.94–1.52) |

All analyses were adjusted for socioeconomic and demographic factors. Each moderating variable was analyzed separately in adjusted models, with mutual adjustment for other school health actions, in order to isolate their specific moderating effects.

[a] PR, prevalence ratio; [b] CI, confidence interval.

* Taking into account the sample weight.

** $p < 0.005$.

Psychosocial adversities, particularly bullying, have been consistently associated with poorer oral health outcomes during adolescence. Exposure to interpersonal violence has been linked to increased oral health conditions through multiple pathways, including heightened psychological distress, impaired self-care behaviors such as oral hygiene, and reduced access to or avoidance of preventive dental services. These mechanisms may act cumulatively over time, exacerbating the burden of untreated oral disease and reinforcing social gradients in oral health. Furthermore, the persistence of these associations across diverse populations highlights the influence of broader structural and contextual determinants that shape adolescents' vulnerability to oral health problems. Understanding these interactions is essential for developing targeted interventions that address both psychosocial factors and oral health disparities [13].

Despite evidence supporting the association between bullying and adverse oral health outcomes, the absence of moderating effects in our findings suggests that, at least in their current form and scope, the school-level interventions under investigation were not sufficiently effective to buffer the impact of bullying or reduce oral health problems among adolescents [13]. It likely reflects the lack of a cohesive and standardized programmatic framework within the School Health Program (PSE). Its conception as an intersectoral public policy bridging health and education constitutes an important advance toward comprehensive health care for children and adolescents [8,13,14]. Nevertheless, the mere presence of the program in schools may not be sufficient to influence complex psychosocial exposures [15].

In addition, in some school environments, there appears to be no clear accountability at the local level for implementing a comprehensive, evidence-based package of interventions targeting bullying prevention, oral health promotion, or related issues. Previous research has demonstrated that the effectiveness of school-based strategies depends on regularity, intersectoral coordination, and responsiveness to social determinants of health specific to each territory [13,16]. These

findings underscore the missed opportunity represented by the PSE and suggest that, in its current form, it may represent an inefficient allocation of resources and funding.

Similarly, no moderation effect was found for oral health promotion or bullying prevention activities. While these strategies are fundamental within school health initiatives, they may not affect outcomes such as toothache if implemented in a fragmented, occasional manner or disconnected from students' subjective experiences [8,14]. School initiatives focused primarily on clinical or informational aspects of oral health tend to overlook broader psychosocial determinants linking interpersonal violence to physical suffering [15]. The adoption of integrated approaches based on the common risk factor framework may enhance the articulation between oral health, mental health, and the promotion of protective school environments, thus supporting more effective and sustainable interventions over time [14,16]. Although the PSE encourages oral health education activities such as workshops, games, supervised toothbrushing, and fluoride application, the effectiveness of these strategies depends on how they are implemented at the local level [17]. The existence of such actions alone does not ensure impact, particularly when they are occasional, fragmented, or disconnected from the school's pedagogical plan and from students' psychosocial context. Evidence indicates that school-based oral health promotion requires sustained and context-sensitive implementation, integration with broader educational strategies, and alignment with social determinants of health to produce equitable and long-term benefits [18].

This study presents limitations that must be acknowledged. First, the cross-sectional design precludes causal inference, as exposure and outcome were assessed simultaneously, making it impossible to establish temporal precedence. Second, although the study employed robust Poisson regression models with standard errors adjusted for clustering at the school level, it was not feasible to estimate multilevel models due to convergence issues and missing data at the contextual level. Additionally, information on oral hygiene practices, consumption of sugary drinks, and mental health (such as depressive symptoms, stress, and sleep disorders) was not available in PeNSE, which were not measured and could influence the associations observed. While the interaction terms included in the models allowed for the investigation of potential moderation effects of school-level variables on the association between bullying and toothache, the absence of random effects modeling restricts the interpretation of these interactions as contextual effects in the strictest sense. Therefore, the findings should be interpreted as population-averaged estimates, rather than school-specific effects. Additionally, reliance on self-reported data may introduce recall or reporting bias, particularly for sensitive topics, such as bullying and pain experiences. Despite these limitations, the study provides important insights into the relationship between school environment and adolescents' oral health experiences.

The central finding of this study is the association between bullying and dental pain, suggesting that bullying may act as a psychosocial mechanism contributing to inequalities in oral health during adolescence. Dental pain serves as a tangible indicator of untreated oral disease, and the social adversity experienced through bullying can exacerbate these conditions by influencing behaviors, stress responses, and access to care. Recognizing bullying as a key social determinant provides an essential perspective for addressing oral health disparities and calls for comprehensive strategies that go beyond clinical interventions to incorporate psychosocial support within the school environment. Policies such as the PSE and oral health promotion initiatives should consider incorporating psychosocial components, ensuring continuity of actions, and fostering school environments that support comprehensive care and health equity.

## Author contributions

**Conceptualization:** Caroline Segatto Girardon, Maria Laura Braccini Fagundes, Fernando Neves Hugo, Orlando Luiz do Amaral Júnior.

**Data curation:** Caroline Segatto Girardon, Orlando Luiz do Amaral Júnior.

**Formal analysis:** Orlando Luiz do Amaral Júnior.

**Investigation:** Caroline Segatto Girardon, Orlando Luiz do Amaral Júnior.

**Methodology:** Caroline Segatto Girardon, Maria Laura Braccini Fagundes, Fernando Neves Hugo, Orlando Luiz do Amaral Júnior.

**Project administration:** Orlando Luiz do Amaral Júnior.

**Supervision:** Fernando Neves Hugo, Jessye Melgarejo do Amaral Giordani, Luana Severo Alves, Orlando Luiz do Amaral Júnior.

**Validation:** Jessye Melgarejo do Amaral Giordani, Orlando Luiz do Amaral Júnior.

**Visualization:** Maria Laura Braccini Fagundes, Fernando Neves Hugo, Jessye Melgarejo do Amaral Giordani, Luana Severo Alves, Orlando Luiz do Amaral Júnior.

**Writing – original draft:** Caroline Segatto Girardon, Maria Laura Braccini Fagundes, Fernando Neves Hugo, Jessye Melgarejo do Amaral Giordani, Luana Severo Alves, Orlando Luiz do Amaral Júnior.

**Writing – review & editing:** Caroline Segatto Girardon, Maria Laura Braccini Fagundes, Fernando Neves Hugo, Jessye Melgarejo do Amaral Giordani, Luana Severo Alves, Orlando Luiz do Amaral Júnior.

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
