## [Decision Letter · Decision Letter 0]

3 Oct 2025

PGPH-D-25-02295

The Association between bullying and toothache in Brazilian students: an analysis of the Brazilian National Student Health Survey

Dear Dr. Hugo,

Thank you for submitting your manuscript to PLOS Global Public Health. After careful consideration, we feel that it has merit but does not fully meet PLOS Global Public Health’s publication criteria as it currently stands. Therefore, we invite you to submit a revised version of the manuscript that addresses the points raised during the review process.

We look forward to receiving your revised manuscript.

Kind regards,

Pengpeng Ye

Academic Editor

Journal Requirements:

1. Please send a completed 'Competing Interests' statement, including any COIs declared by your co-authors. If you have no competing interests to declare, please state "The authors have declared that no competing interests exist". Otherwise please declare all competing interests beginning with the statement "I have read the journal's policy and the authors of this manuscript have the following competing interests:"

1. Please clarify all sources of funding (financial or material support) for your study. List the grants (with grant number) or organizations (with url) that supported your study, including funding received from your institution.

2. State the initials, alongside each funding source, of each author to receive each grant.

3. State what role the funders took in the study. If the funders had no role in your study, please state: “The funders had no role in study design, data collection and analysis, decision to publish, or preparation of the manuscript.”

4. If any authors received a salary from any of your funders, please state which authors and which funders.

Reviewers' comments:

Reviewer's Responses to Questions

**Comments to the Author**

1. Does this manuscript meet PLOS Global Public Health’s publication criteria? Is the manuscript technically sound, and do the data support the conclusions? The manuscript must describe methodologically and ethically rigorous research with conclusions that are appropriately drawn based on the data presented.? Is the manuscript technically sound, and do the data support the conclusions? The manuscript must describe methodologically and ethically rigorous research with conclusions that are appropriately drawn based on the data presented.

Reviewer #1: Yes

Reviewer #2: Yes

2. Has the statistical analysis been performed appropriately and rigorously?

Reviewer #1: Yes

Reviewer #2: Yes

3. Have the authors made all data underlying the findings in their manuscript fully available (please refer to the Data Availability Statement at the start of the manuscript PDF file)?

The PLOS Data policy requires authors to make all data underlying the findings described in their manuscript fully available without restriction, with rare exception. The data should be provided as part of the manuscript or its supporting information, or deposited to a public repository. For example, in addition to summary statistics, the data points behind means, medians and variance measures should be available. If there are restrictions on publicly sharing data—e.g. participant privacy or use of data from a third party—those must be specified.requires authors to make all data underlying the findings described in their manuscript fully available without restriction, with rare exception. The data should be provided as part of the manuscript or its supporting information, or deposited to a public repository. For example, in addition to summary statistics, the data points behind means, medians and variance measures should be available. If there are restrictions on publicly sharing data—e.g. participant privacy or use of data from a third party—those must be specified.

Reviewer #1: No

Reviewer #2: Yes

4. Is the manuscript presented in an intelligible fashion and written in standard English?

Reviewer #1: Yes

Reviewer #2: Yes

Reviewer #1: The topic is timely and relevant for adolescent oral health and school health policy in LMICs, and the use of a large, nationally representative dataset is a strength. However, minor revision is needed as per the following comments.

1) Please ensure that all in-text citations are placed before the full stop.

2) It is observed that toothache was slightly higher for students bullied once than ≥2 times, please briefly explain this pattern in the Discussion.

3) Consideration of Confounding: If available in PeNSE, consider additional proxies (e.g., oral hygiene/sugary beverages, mental health/stress, richer SES). If not available, state this as a limitation.

4) The underlying PeNSE data are public, but readers need a direct data link, variable list, and analysis code (cleaning/model syntax)

5) I would suggest adding a main figure related to adjusted predicted prevalence of toothache by bullying frequency (none/once/≥2).

6) Also, the authors can consider another figure that can be useful related to the moderation plot showing predicted prevalence by bullying frequency, stratified by each school action (Yes/No).

Reviewer #2: The topic is timely and policy relevant, linking psychosocial exposures to oral-health outcomes in adolescence is valuable. However, several methodological choices (confounder selection, handling of the complex survey, missingness strategy, and interpretation of “moderation”) limit causal and policy inferences. Effect sizes are small and highly vulnerable to residual bias. With revisions, the paper could offer a cautious, descriptive contribution and a stronger platform for future quasi-experimental work on PSE and related school interventions.

Confounding: The adjusted PRs (1.07 – 1.08) are much smaller than crude prevalence differences (eg 32.5% vs 21.5% toothache, “once” vs “never”), suggesting model specification strongly drives estimates. At minimum, provide a causal DAG to justify covariates. In particular, recent dental-service use is likely a mediator/collider (toothache leads to care seeking; bullying could also reduce/raise care seeking), so adjusting for it may bias estimates toward the null. Reestimate models without this variable and with alternative confounder sets (age, sex, SES proxies, region/UF, urban/rural) and add mental-health controls (depressive symptoms, stress, sleep) and oral-health behaviors (sugary drinks, hygiene frequency) available in PeNSE to reduce residual confounding.

Weights: The text and tables note weights, but it’s unclear if strata and primary sampling units were incorporated via svyset in Stata. Clustering on school alone does not fully account for PeNSE’s multi-stage design. Please report your svy settings (weight, PSU, strata variables) and re-run models with svy: poisson (or svy: glm, family(poisson) link(log) vce(robust)), including for interaction models.

Missingness: Restricting to complete cases (final n=53,711 from an initial 159,245) risks selection bias if missingness is not MCAR. Provide a CONSORT/STROBE-style flow, proportions missing by variable, and comparisons of included vs excluded students. Consider multiple imputation (MICE) compatible with survey design (impute within strata/PSUs or include design variables in the imputation) and present imputed-data estimates alongside complete-case results. Your rationale about multilevel convergence is noted, but MI does not require multilevel outcome modeling.

Effect size: Adjusted PRs of 1.07–1.08 are statistically significant largely due to sample size as the E-value for 1.08 is small, meaning modest unmeasured confounding could explain the association. Provide E-values and absolute risk differences or marginal predicted probabilities to aid clinical/health-economics interpretation. Explore dose response (treat bullying frequency ordinally and test linear trend) and present margins plots. The non-monotonic raw prevalence (once > twice+) deserves comment and sensitivity checks (e.g., alternative categorizations). If bullying truly increases toothache risk in a dose–response way, you’d expect prevalence to rise as frequency rises: Never < Once < Twice+. When the crude data show Once > Twice+, that breaks monotonicity. It doesn’t prove the association is spurious, but it’s a red flag that measurement, confounding, or selection might be shaping the pattern.

In addition, the three school-level moderators are binary presence/absence and likely implementation-insensitive. Null interactions may reflect low fidelity/intensity variation, not true absence of effect. If feasible, construct school-level indices (e.g., number of health actions, frequency, integration into pedagogy) or aggregate multiple items to reduce measurement error. Present ICC for toothache by school, and consider state/UF fixed effects to absorb contextual confounding.

**Do you want your identity to be public for this peer review?** For information about this choice, including consent withdrawal, please see our Privacy Policy..

Reviewer #1: No

Reviewer #2: No

---

## [Decision Letter · Decision Letter 1]

30 Dec 2025

PGPH-D-25-02295R1

The Association between bullying and toothache in Brazilian students: an analysis of the Brazilian National Student Health Survey

Dear Dr. Hugo,

Thank you for submitting your manuscript to PLOS Global Public Health. After careful consideration, we feel that it has merit but does not fully meet PLOS Global Public Health’s publication criteria as it currently stands. Therefore, we invite you to submit a revised version of the manuscript that addresses the points raised during the review process.

We look forward to receiving your revised manuscript.

Kind regards,

Pengpeng Ye

Academic Editor

Journal Requirements:

Additional Editor Comments (if provided):

Reviewers' comments:

Reviewer's Responses to Questions

**Comments to the Author**

Reviewer #1: All comments have been addressed

Reviewer #2: All comments have been addressed

publication criteria? Is the manuscript technically sound, and do the data support the conclusions? The manuscript must describe methodologically and ethically rigorous research with conclusions that are appropriately drawn based on the data presented.? Is the manuscript technically sound, and do the data support the conclusions? The manuscript must describe methodologically and ethically rigorous research with conclusions that are appropriately drawn based on the data presented.

Reviewer #1: Yes

Reviewer #2: Yes

3. Has the statistical analysis been performed appropriately and rigorously?

Reviewer #1: Yes

Reviewer #2: Yes

4. Have the authors made all data underlying the findings in their manuscript fully available (please refer to the Data Availability Statement at the start of the manuscript PDF file)?

The PLOS Data policy requires authors to make all data underlying the findings described in their manuscript fully available without restriction, with rare exception. The data should be provided as part of the manuscript or its supporting information, or deposited to a public repository. For example, in addition to summary statistics, the data points behind means, medians and variance measures should be available. If there are restrictions on publicly sharing data—e.g. participant privacy or use of data from a third party—those must be specified.requires authors to make all data underlying the findings described in their manuscript fully available without restriction, with rare exception. The data should be provided as part of the manuscript or its supporting information, or deposited to a public repository. For example, in addition to summary statistics, the data points behind means, medians and variance measures should be available. If there are restrictions on publicly sharing data—e.g. participant privacy or use of data from a third party—those must be specified.

Reviewer #1: Yes

Reviewer #2: Yes

5. Is the manuscript presented in an intelligible fashion and written in standard English?

Reviewer #1: Yes

Reviewer #2: Yes

Reviewer #1: The author has adequately responded to all comments and queries, providing substantial revisions that have significantly improved the clarity and rigor of the work

Reviewer #2: Thank you for the revision and detailed response to Round 1. The manuscript is meaningfully improved and several core methodological concerns have been addressed, notably: (i) the inclusion of a causal framework (DAG) clarifying which variables are treated as confounders versus potential mediators, (ii) corresponding adjustments to the covariate strategy to avoid overadjustment, and (iii) the addition of predicted/adjusted prevalence figures that substantially improve interpretability relative to reporting only prevalence ratios.

That said, I still see one major issue and a few important transparency items that should be resolved before the paper is fully publication-ready:

Major: internal inconsistency in the stated estimation approach for the complex survey design. The revised methods indicate analyses were conducted using Stata’s survey procedures with svy and incorporation of weights/PSUs/strata, but the Statistical Analysis section also discusses the use of robust variance estimation with standard errors clustered at the school level as an “alternative strategy.” These are not equivalent approaches, and the manuscript currently reads as if both are used without a clear statement of which is the primary specification and which (if any) is a sensitivity check. This needs to be made internally consistent: either (a) present the survey-design-based estimation as the main approach and remove/relegate the cluster-robust discussion to sensitivity analysis, or (b) if survey estimation was not ultimately used, revise the text accordingly and justify the final choice.

Transparency: complete-case restriction remains under-documented. While the revised text provides a clearer narrative rationale for complete-case analysis, it still omits the diagnostic information requested in Round 1: missingness rates by variable, a simple sample flow/attrition summary, and a comparison of included vs excluded observations on key demographics/SES markers. Given the large reduction from the initial sample to the analytic sample, a short appendix table would materially strengthen credibility and allow readers to assess potential selection issues.

Moderation measures remain coarse. The revision more clearly acknowledges the limitations of binary indicators for school actions and helps interpret the null interaction results. However, the analysis would benefit from at least minimal additional reporting (e.g., ICC or other clustering diagnostics, if invoked) and/or clearer framing that the moderator measures capture presence/absence rather than intensity/implementation quality.

Overall, I view the paper as substantially improved and close, but I recommend the editor request a targeted revision to (i) fully align and clearly state the survey/variance estimation strategy and (ii) add basic missingness/sample-selection diagnostics (ideally in an appendix). These are feasible changes that would materially improve reproducibility and confidence in the results.

**Do you want your identity to be public for this peer review?** For information about this choice, including consent withdrawal, please see our Privacy Policy..

Reviewer #1: No

Reviewer #2: No

---

## [Decision Letter · Decision Letter 2]

13 Mar 2026

The Association between bullying and toothache in Brazilian students: an analysis of the Brazilian National Student Health Survey

PGPH-D-25-02295R2

Dear Dr. Hugo,

We are pleased to inform you that your manuscript 'The Association between bullying and toothache in Brazilian students: an analysis of the Brazilian National Student Health Survey' has been provisionally accepted for publication in PLOS Global Public Health.

Best regards,

Pengpeng Ye

Academic Editor

Reviewer Comments (if any, and for reference):

Reviewer's Responses to Questions

**Comments to the Author**

Reviewer #2: All comments have been addressed

publication criteria? Is the manuscript technically sound, and do the data support the conclusions? The manuscript must describe methodologically and ethically rigorous research with conclusions that are appropriately drawn based on the data presented.? Is the manuscript technically sound, and do the data support the conclusions? The manuscript must describe methodologically and ethically rigorous research with conclusions that are appropriately drawn based on the data presented.

Reviewer #2: Yes

3. Has the statistical analysis been performed appropriately and rigorously?

Reviewer #2: Yes

4. Have the authors made all data underlying the findings in their manuscript fully available (please refer to the Data Availability Statement at the start of the manuscript PDF file)?

The PLOS Data policy requires authors to make all data underlying the findings described in their manuscript fully available without restriction, with rare exception. The data should be provided as part of the manuscript or its supporting information, or deposited to a public repository. For example, in addition to summary statistics, the data points behind means, medians and variance measures should be available. If there are restrictions on publicly sharing data—e.g. participant privacy or use of data from a third party—those must be specified.requires authors to make all data underlying the findings described in their manuscript fully available without restriction, with rare exception. The data should be provided as part of the manuscript or its supporting information, or deposited to a public repository. For example, in addition to summary statistics, the data points behind means, medians and variance measures should be available. If there are restrictions on publicly sharing data—e.g. participant privacy or use of data from a third party—those must be specified.

Reviewer #2: Yes

5. Is the manuscript presented in an intelligible fashion and written in standard English?

Reviewer #2: Yes

Reviewer #2: The author has adequately responded to all comments and queries, providing substantial revisions that have significantly improved the clarity and rigor of the work.

**Do you want your identity to be public for this peer review?** For information about this choice, including consent withdrawal, please see our Privacy Policy..

Reviewer #2: No
